# Plastid Molecular Chaperone HSP90C Interacts with the SecA1 Subunit of Sec Translocase for Thylakoid Protein Transport

**DOI:** 10.3390/plants13091265

**Published:** 2024-05-01

**Authors:** Adheip Monikantan Nair, Tim Jiang, Bona Mu, Rongmin Zhao

**Affiliations:** Department of Biological Sciences, University of Toronto Scarborough, Toronto, ON M1C 1A4, Canada; Department of Cell & Systems Biology, University of Toronto, Toronto, ON M5S 3B2, Canada; adheip.nair@mail.utoronto.ca (A.M.N.); tim.jiang@mail.utoronto.ca (T.J.); bona.mu@mail.utoronto.ca (B.M.)

**Keywords:** HSP90C, SecA1, protein translocation, chloroplast proteostasis, thylakoid membrane

## Abstract

The plastid stroma-localized chaperone HSP90C plays a crucial role in maintaining optimal proteostasis within chloroplasts and participates in protein translocation processes. While existing studies have revealed HSP90C’s direct interaction with the Sec translocase-dependent client pre-protein PsbO1 and the SecY1 subunit of the thylakoid membrane-bound Sec1 translocase channel system, its direct involvement with the extrinsic homodimeric Sec translocase subunit, SecA1, remains elusive. Employing bimolecular fluorescence complementation (BiFC) assay and other in vitro analyses, we unraveled potential interactions between HSP90C and SecA1. Our investigation revealed dynamic interactions between HSP90C and SecA1 at the thylakoid membrane and stroma. The thylakoid membrane localization of this interaction was contingent upon active HSP90C ATPase activity, whereas their stromal interaction was associated with active SecA1 ATPase activity. Furthermore, we observed a direct interaction between these two proteins by analyzing their ATP hydrolysis activities, and their interaction likely impacts their respective functional cycles. Additionally, using PsbO1, a model Sec translocase client pre-protein, we studied the intricacies of HSP90C’s possible involvement in pre-protein translocation via the Sec1 system in chloroplasts. The results suggest a complex nature of the HSP90C-SecA1 interaction, possibly mediated by the Sec client protein. Our studies shed light on the nuanced aspects of HSP90C’s engagement in orchestrating pre-protein translocation, and we propose a potential collaborative role of HSP90C with SecA1 in actively facilitating pre-protein transport across the thylakoid membrane.

## 1. Introduction

Heat shock proteins (HSPs) represent evolutionarily conserved molecular chaperones that provide resistance against stress, notably temperature stressors, across diverse organisms [1,2,3]. Beyond their role in stress response, HSPs fulfill critical functions in cellular integrity maintenance, signal transduction, and protein translocation processes [1,4]. Among these, heat shock protein 90 (HSP90), a class of highly conserved molecular chaperones weighing approximately 90 kDa, constitutes approximately 1–2% of the cellular protein content in unstressed organisms [5,6]. Remarkably, under stress conditions, this proportion could escalate to around 5% of the total cellular protein content [7,8]. In yeast, HSP90 is functionally associated with nearly 10% of all yeast proteomes and ~20% of all genes, underscoring its abundance and extensive involvement across various cellular processes [9,10].

Chloroplasts, a specialized type of plastid, are prominent organelles responsible for light absorption and high-energy sugar molecule synthesis through photosynthesis in higher plants. Thylakoid membranes and chlorophyll pigments are unique to chloroplasts and essential for their photosynthetic function. In the absence of light, the developmental progression of undifferentiated proplastids halts at an intermediate stage known as the etioplast [11,12]. On the other hand, light-induced damage during photosynthesis leads to photooxidative stress due to light-induced reactive oxygen species generation [13]. Additionally, heat-induced damage could disrupt the photochemical reactions within the thylakoid lamellae, resulting in protein aggregation and denaturation [14]. It is, therefore, a persistent challenge for chloroplasts to maintain appropriate protein homeostasis due to the consistent and dynamic adverse impacts on the organelles within plant cells [15]. To navigate these complexities, chloroplasts rely on heat shock proteins, among which the HSP90C subfamily, encompassing HSP90 orthologs in all photosynthetic-capable organisms, assumes a pivotal role in chloroplast biogenesis, development, and pre-protein translocation processes [16,17].

Although chloroplasts harbor their own genetic materials, they heavily rely on nuclear-encoded proteins for proper function. Nuclear-encoded pre-proteins destined for the thylakoid lumen traverse the outer chloroplast membrane via the TOC complex and the inner chloroplast membrane via the TIC complex. Together, these translocons form a supercomplex with channel proteins at the outer and inner envelope membranes’ junctions (OEM/IEM), facilitating pre-protein translocation to the stroma [18]. Energy is required for pre-protein translocation, a process facilitated by the Ycf2 complex involving FtsH-like ATPases and HSPs such as HSP93, cpHSP70, and HSP90C, functioning as pre-protein import motors to transport pre-proteins into the stroma [19,20,21,22]. For those proteins to be targetted to the thylakoid, once traversing the OEM and IEM, they continue their journey towards the thylakoid membrane or the lumen, utilizing distinct translocation pathways, e.g., cpSec, cpTat, and cpSRP [23].

Previous studies have highlighted the direct interaction between HSP90C and the unfolded stromal intermediate form of PsbO1 (*i*PsbO1), a lumen-localized photosystem II subunit responsible for stabilizing manganese clusters crucial for the water oxidation step [24,25]. HSP90C also directly interacts with the membrane-bound SecY1 subunit of the Sec translocase system [26]. PsbO1, a recognized SEC translocase client protein, maintains its unfolded state with HSP90C assistance [24,25,27]. For successful transport, SEC client proteins, especially those bearing thylakoid targeting peptides, necessitate recognition and continuous threading into the channel by the SEC translocon’s extrinsic motor protein, SecA. This mechanism likely follows a catch-and-release model akin to one of many hypothesized bacterial SEC translocation mechanisms [28]. In bacterial systems, the functional cycle initiates with dimeric SecA in an ADP-bound autoinhibited state, which primes itself for SecYEG channel binding by interacting with anionic phospholipids on the bilayer (membrane assembly stage) [29,30]. Subsequently, SecA asymmetrically binds to the SecYEG channel, leading to monomerization and a subsequent ~10-fold increase in pre-protein binding affinity (translocase priming stage) [29,31]. Upon SecA ATPase docking, the pre-protein stabilization is facilitated by the ATP-independent homotetrameric SecB chaperone, known for its robust anti-folding activity, preventing premature folding of pre-proteins [32]. During Sec client protein transport, the pre-protein’s signal peptide engages various sites on SecA, bridging the modules and orchestrating their dynamics to form a “pre-initiation complex” (catch stage). This complex instigates nucleotide cycling, enabling ATP-ADP exchange while allowing pre-protein diffusion through the channel (release stage). Under ATP hydrolysis, SecA undergoes a substantial conformational alteration, particularly in its pre-protein binding domain (PBD) topology [28,33].

Interestingly, while a SecB homolog has not been identified in plastids, it is hypothesized that HSP90C might, to some extent, safeguard Sec clients and facilitate their Sec-dependent translocation [26]. Moreover, unlike bacterial SecA, plastid SecA comprises two homologs, SecA1 (thylakoid membrane-localized) and SecA2 (IEM-localized), each with distinct functions [34]. Knockout of SecA1 results in the albino or glassy yellow (*agy1*) mutants, which show high sensitivity to light conditions and are crucial for chloroplast biogenesis and the light/dark adaptation of the chloroplast [35]. SecA2 mutants, however, consistently yielded embryos arrested at the globular stage, similar to that of SecY2 mutants [36]. Previous works have identified thylakoidal processing peptidase, plastidic type I signal peptidase I (Plsp1), to co-migrate with plastid chaperonin Cpn60 to facilitate membrane insertion through the Sec1 system [37]. They have also established the synergistic efforts between SecA1 and Cpn60 for properly inserting Plsp1 using the Sec1 system, laying the groundwork for plastid chaperone and SecA1 cooperativity.

This study aims to understand other mechanisms of the thylakoid pre-protein translocation by exploring the interaction between *Arabidopsis* HSP90C and SecA1 to elucidate HSP90C’s role in pre-protein translocation. We present evidence suggesting a potential physical association between HSP90C and SecA1 in vivo. Additionally, we conducted in vitro analyses to identify a highly dynamic direct interaction between HSP90C and SecA1. Furthermore, we used PsbO1 as the model Sec client protein to study the role of the client protein in facilitating the interaction. Our data suggest that while HSP90C may interact directly with SecA1, the presence of a client protein changes their interaction status, and we propose a functional model that illustrates the dynamic relationship between HSP90C and SecA1 during pre-protein translocation.

## 2. Results

### 2.1. HSP90C Facilitates an Interaction with the SecA1 ATPase Motor Protein In Vivo

To understand HSP90C’s localization and implication in facilitating the translocation mechanism of Sec client proteins across the thylakoid membrane, we employed a bimolecular fluorescence complementation (BiFC) assay in tobacco (*N. benthamiana*) leaf mesophyll chloroplasts. To establish a baseline, we examined fluorescence patterns through the transient co-expression of half-c/nYFP SecA1 or half-n/cYFP fused to PsbO1^1–58^ or Rubisco small subunit 3B (RbcS3B) as control pairs. As expected, our observations revealed a lack of discernible signals in the PsbO1^1–58^-PsbO1^1–85^, PsbO1^1–85^-RbcS3B, SecA1-RbcS3B, and SecA1-PsbO1^1–58^ YFP co-expression pairs (Figure 1A), implying the absence of self-assembly of the YFP halves in mesophyll chloroplasts and SecA1 does not interact with half of the YFP. Particularly, we observed no fluorescence signal between the PsbO1^1–85^-PsbO1^1–58^ or PsbO1^1–85^-PsbO1^1–58^ YFP co-expression pairs, indicating the PsbO1 thylakoid targeting peptide does not facilitate interaction with either half of the YFP protein. We did observe interactions between HSP90C and PsbO1^1–85^ (Figure 1A) as previously reported [26]. Representative immunoblot analyses validated the successful transient expression of all YFP fusion proteins (Appendix A).

Interestingly, the pairing of SecA1 fusion with the half-YFP fusion from PsbO1^1–332^ resulted in distinct fluorescence signals (Figure 1B). Co-expression with the half-YFP fusion from PsbO1^1–85^ yielded YFP signals with intensities on par with the SecA1-PsbO1^1–332^ co-expression line. This observation strongly implies a robust interaction between SecA1 and the PsbO1 thylakoid targeting sequence alone within chloroplasts. However, the discernible distinction lies in the spatial distribution of the YFP fluorescence signals. Notably, the green YFP fluorescence signal extensively coincides with the chlorophyll autofluorescence signal originating from the thylakoid membrane in the SecA1-PsbO1^1–85^ YFP co-expression pair. Conversely, upon co-expressing SecA1 and the full-length PsbO1 (PsbO1^1–332^) YFP fusion pairs, we observe YFP signals that exhibit a reduced overlap with chlorophyll autofluorescence signals, which strongly suggests that the interaction between SecA1 and the Sec client protein may predominantly occur within the stromal environment. Moreover, our investigation reaffirmed the previously reported interaction between SecA1 and SecY1 (Figure 1B) [38]. Importantly, our findings reveal a substantial interaction between SecA1 and HSP90C, characterized by a prominent YFP fluorescence signal (Figure 1B) that overlaps with chlorophyll autofluorescence signals and is present in chlorophyll-poor regions, indicating the spatial preference of this interaction towards the stroma as well as the thylakoid membrane.

We further investigated the spatial dynamics of SecA1 interactions within distinct chloroplast compartments, specifically, the chlorophyll-rich thylakoids and the chlorophyll-poor stroma. We examined the distribution of YFP signal localization within individual mesophyll chloroplasts treated with the HSP90C inhibitor geldanamycin and SecA1 inhibitor. We first observed chloroplast stroma-localized interaction for SecA1-SecA1, HSP90C-PsbO1^1–85^, SecA1-PsbO1^1–85^, and SecA1-PsbO1^1–332^ YFP co-expression pairs (Figure 2A). We observed strong thylakoid membrane localization in the SecA1-SecY1 YFP co-expression line and dual localization towards the thylakoid membrane and stroma for the SecA1-HSP90C YFP co-expression line (Figure 2A). As previously documented, mesophyll chloroplasts were subjected to 10 mM sodium azide treatment to inhibit SecA1’s ATPase activity, thereby impeding its translocation function [39]. Incubation with sodium azide resulted in a discernible decrease in YFP signals within the co-expression lines of SecA1-SecA1 and SecA1-SecY1, indicative of the efficacy of sodium azide as an agent impacting these interactions (Figure 2B, left column). Intriguingly, an intensified YFP signal was detected between HSP90C and the PsbO1^1–85^ thylakoid signaling sequence, consistent with prior observations [26]. Surprisingly, decreased YFP signals were evident between SecA1 and PsbO1^1–85^/PsbO1^1–332^ fusion protein pairs within the chlorophyll-poor regions, implying a diminished stromal interaction (Figure 2B, left column). The observed reduction in the stromal interaction between SecA1 and HSP90C suggests a dependence on active SecA1 for facilitating the interaction with HSP90C in the stroma.

In contrast, the inhibition of HSP90C activity using geldanamycin resulted in an amplified YFP signal primarily within the chlorophyll-poor stroma regions in the HSP90C and SecA1-PsbO1^1–85^ YFP co-expression pairs (Figure 2B, right column). No drastic change in the YFP signal localization was observed in the SecA1 and SecA1/SecY1/PsbO1^1–332^ YFP co-expression pairs (Figure 2B, right column). Interestingly, geldanamycin facilitated the dissociation of SecA1-HSP90C YFP interaction from the thylakoid membrane while preserving the stromal interaction, demonstrated by a significant reduction in the overlap between the YFP and chlorophyll autofluorescence signals. This observation suggests that the maintenance of the interaction between HSP90C and SecA1 at the thylakoid membrane requires the involvement of active HSP90C.

### 2.2. The Sec Client Protein May Mediate the HSP90C and SecA1 Interaction

Given the dynamic nature of the SecA1 and HSP90C interaction, alongside their ATPase activity-dependent localization, we attempted to visualize their interaction through other established biochemical techniques. Initial attempts using yeast-two hybrid and in vitro pulldown assays involving His_6_-tagged proteins expressed and purified from *E. coli* failed to reliably demonstrate interactions compared to control conditions (Appendix A). This led us to hypothesize that their interaction might occur transiently in vivo, possibly during specific stages of client protein binding and translocation, or be contingent upon distinct conformations of both ATPase proteins, given their ATP-hydrolysis-driven conformational changes [33,40,41,42]. We sought to investigate the potential interaction between HSP90C and SecA1 using in vitro size exclusion chromatography (SEC), both in the absence and presence of ATP. However, our SEC analysis did not reveal a discernible interaction between HSP90C and SecA1 (Appendix A). We confirmed and verified HSP90C chaperone activity by observing the prevention of thermal-induced aggregation of citrate synthase and measuring the chaperone’s ATPase activity (Appendix A). We verified that HSP90C’s ATP-independent chaperone activity is dosage-dependent and observed a maximal ATP hydrolysis rate (V_max_) of 18.17 pmol/min with an ATP binding affinity (K_M_) value of 4.01 mM.

Expanding our investigation, we introduced a mutant variant of the client protein PsbO1 into the reaction mixture. This variant, distinguished by a point mutation at the 200th amino acid that replaces threonine with alanine, is previously recognized for its enhanced affinity to HSP90C [24]. Remarkably, the inclusion of PsbO1 led to noticeable shifts in the elution profiles of SecA1 when combined with HSP90C and PsbO1, suggesting the formation of a complex (Figure 3). This observation supports the plausibility that SecA1 and HSP90C form a guiding complex with the Sec client protein, potentially facilitating the transition of the client protein from a chaperone-stabilized state to one stabilized by SecA1.

Our pulldown assays and SEC analyses were conducted at a low temperature (4 °C), mitigating ATP hydrolysis and preventing dynamic conformational changes in both HSP90C and SecA1. Under these conditions, the detection of interactions reliant on specific intermediate conformations might be unfavorable. To assess ATP hydrolysis at normal or elevated temperatures, we measured the ATPase activities of HSP90C and SecA1. Utilizing NADH-coupled spectroscopic analysis [43], we investigated their basal ATP turnover rates at a saturated 5 mM ATP, approximating maximum ATP hydrolysis turnover rates. Notably, the turnover rates for HSP90C and SecA1 were determined to be 0.880 ± 0.074 min^−1^ and 2.046 ± 0.124 min^−1^, respectively (Figure 4). Interestingly, the combined turnover rate of both SecA1 and HSP90C, measured at 3.488 ± 0.100 min^−1^, deviated from the sum of the individual proteins’ average turnover rates of 2.93 min^−1^. This substantial increase in the apparent ATP turnover rate strongly indicates a physical interaction between HSP90C and SecA1, resulting in the stimulation of one or both ATP hydrolysis cycles. The turnover rate upon the inclusion of SecA1, HSP90C, and PsbO1 shows a discernible 16% escalation relative to the SecA1 + HSP90C mixture (Figure 4). The observed modulation in turnover rates provides further support for the notion of an interaction interface between SecA1 and HSP90C facilitated by the presence of the PsbO1 client protein. Furthermore, this ternary complex is hypothesized to serve as a prelude to subsequent translocation channel priming.

## 3. Discussion

Recent investigations have underscored the potential role of HSP90C in facilitating pre-protein translocation via the chloroplastic Sec translocase system [26]. The intricate chloroplastic Sec1 system comprises SecY1, SecE1, and the core extrinsic SecA1 subunit orchestrating the active transport of the pre-protein translocation [44]. Comparatively to bacterial orthologs (SecYEG complex) and eukaryotic endoplasmic reticulum counterparts (Sec61αβγ system), the plastid Sec1 translocase remains relatively unexplored. This study examined the dynamic interplay between HSP90C and SecA1, both in vivo and in vitro. Our initial observations using the BiFC assays in *Nicotiana benthamiana* unveiled robust interactions among SecA1, HSP90C, and the Sec client protein PsbO1. The dynamic interactions between HSP90C and SecA1 were localized at both the thylakoid membrane and the stroma (Figure 1 and Figure 2). Intriguingly, the localization of the SecA1-HSP90C interaction was also dependent on the activity of both proteins and inhibition of their ATPase activity seemed to alter their in vivo interaction location (Figure 2B). When purified SecA1 and HSP90C were combined, we also observed a significant increase in the combined turnover rate of ATP, suggesting their direct interaction and a stimulatory influence on each other (Figure 4). Their interaction is suggested to be further exemplified in the presence of Sec client protein PsbO1 (Figure 3 and Figure 4). Collectively, based on previous studies on how SecA may bind the clients and associate with the thylakoid membrane channel protein SecYE [28,29,30], we proposed a model (Figure 5) to outline specific HSP90C-SecA1 interactions and how HSP90C may be involved in the plastid Sec translocase functional cycle. Following the HSP90C-mediated stabilization of the Sec client protein, two plausible pathways emerge. The binary HSP90C-Sec client complex might interact with dimeric SecA1, forming a ternary complex targeting the SecYE channel, with findings corroborated by SEC analyses (Figure 3). Alternatively, this binary complex could target SecA1 pre-primed and bound on the SecYE channel.

### 3.1. The Thylakoid Signaling Peptide Sequence Is Sufficient for cpSecA1 Recruitment

The functional significance of the thylakoid signaling peptide sequence in the recruitment of SecA1 is pivotal in chloroplast protein translocation. Our BiFC assay distinctly showcased fluorescence signals present upon transient expression of the SecA1-PsbO1^1–85^ YFP pair, contrasting with the absence of signals in the SecA1-PsbO1^1–58^ YFP pair that consisted only of the chloroplast targeting peptide. The signal strength found in the SecA1-PsbO1^1–85^ YFP line is akin to that of the SecA1-PsbO1^1–332^ co-expression line, indicating the significance of this sequence in the interaction initiation (Figure 1B and Figure 2A). Furthermore, our investigation revealed that the chloroplast targeting peptide (cTP) lacked preferential binding to SecA1, consistent with stromal processing peptidases’ action on chloroplast targeting sequences upon stroma entry, leaving the thylakoid signaling peptide (tSP) exposed [45]. This was further exemplified with our size exclusion chromatography assay, which utilized a mature form of PsbO1 devoid of the thylakoid signaling peptide and exhibited no prominent change in the elution profile of SecA1 or PsbO1 when mixed in equal amounts together. This observation showed that the absence of the tSP on the PsbO1 protein could not initiate the direct interaction with SecA1. Assuming the conservation of the functional cycle between plastidic and bacterial SecA, the thylakoid signaling peptide would assume significance in the initial binding to SecA1 and initiating its ATPase cycling. The presence of this peptide likely primes SecA1’s initial pre-protein binding and initiates ATP cycling, similar to bacterial SecA’s mechanism relying on the first signal sequence for “clamp” closure and subsequent ATP cycling [28].

Bacterial SecA has been reported to use at least 10 piconewtons of mechanical force as a mechanical unfoldase to destabilize folded proteins, which is comparable to other cellular unfoldases [46]. With *m*PsbO1, SecA1 must depend on signal sequences beyond the tSP, which is confined to an α-helical structure. Thus, SecA1 would likely have to actively unfold the first α-helix structure to initiate client binding and trigger ATPase cycling, which may not be preferential. This mechanical force likely stems from the conformational changes that arise from ATP hydrolysis, though the exact mechanism remains to be investigated for plastid SecA1.

Furthermore, the predominantly stromal interaction between SecA1 and PsbO1 raises questions about SecA1’s oligomeric state and its dynamics on client protein binding. Considering SecA1’s stromal localization as a dimer, potentially autoinhibited, our size exclusion chromatography analysis utilized monomeric SecA1 to probe its interaction with PsbO1. Investigating the impact of dimeric SecA1 on client binding stands as a pertinent avenue for further exploration.

### 3.2. HSP90C and SecA1 May Interact through a Complex Established at the SecYE Channel or Form a Guiding Complex Mediated by the Sec Client Protein

Our in vivo interaction analyses (Figure 1 and Figure 2) underscore the interaction dynamics between HSP90C and SecA1, localizing both at the stroma and the thylakoid membrane. Fluorescent signals from SecA1-PsbO1 and SecA1-HSP90C YFP co-expression lines confirm the presence of both stromal and membrane interactions. Inhibition of HSP90C’s ATPase activity emphasizes heightened interaction localization at the stroma. HSP90C’s ATP-independent client binding capability suggests a plausible formation of a guiding complex with SecA1 in the stroma initially, subsequently targeting the SecYE channel, a process in which the client thylakoid targeting sequence may play a minimal role for the SecYE targeting. Our observation of consistent HSP90C and SecA1 interaction at the thylakoid membrane, which requires active HSP90C, supports the presence of this process. As another piece of evidence, previous works have shown that HSP90C and the SecY1 subunit interact directly with each other [26]. However, the exact collaborative role of HSP90C and SecA1 in protein translocation through the SecYE channel remains to be fully elucidated.

Conversely, upon inhibiting SecA1’s ATPase activity, we also observed increased interaction between HSP90C and SecA1 at the thylakoid membrane and reduced interaction at the stroma. Sodium azide, known for disrupting bacterial SecA’s metal binding domain (MBD) and affinity to phospholipids, interestingly accentuated HSP90C-SecA1 interaction at the thylakoid membrane [47,48]. The highly conserved MBD coordinates one Zn^2+^ ion by the CXCX_8_CH motif at the C-terminal tail in the bacterial SecA [47]. Additionally, the MBD is necessary to recruit SecB and the 70S ribosome in the bacterial systems [49,50]. However, this motif is not conserved in plastid SecA1, which raises questions about the existence of the MBD in plastid SecA isoforms and the efficacy of sodium azide as a potential inhibitor of SecA in plant systems. Nevertheless, we see an interaction take place between SecA1 and HSP90C. The interaction between SecA1 and HSP90C may not occur in the same mechanism as that of SecA-SecB in bacterial systems. Our in vivo and in vitro analyses suggest that HSP90C and SecA1’s ATPase activities exert critical functional influences directly on each other through unknown mechanisms (Figure 2 and Figure 4). SecB stimulates SecA ATPase activity in bacterial systems by ~35% [51]. Our findings suggest a synergistic modulation of ATPase activity exhibited by either SecA1 or HSP90C in the presence of each other. Moreover, investigating the difference in the stimulation of SecA1 and HSP90C specifically, along with delineating the degree of difference in activity, would help refine our mechanistic hypotheses. However, this necessitates the availability of SecA1-specific ATPase inhibitors possessing efficacy comparable to radicicol towards HSP90, which exhibits a potent nanomolar affinity [52,53,54]. Presently, to our knowledge, the inhibitors of this nature remain elusive. It is noteworthy that, despite its previous use as a classical SecA inhibitor, sodium azide primarily functions as a SecA-SecYEG inhibitor in bacterial systems [55].

Regardless, our studies together imply that HSP90C and SecA1 interact at the thylakoid membrane, likely with the formation of a supercomplex between SecA1-PsbO1-HSP90C-SecY1. We also provide evidence that shows enhanced binding and elevated ATPase activity of HSP90C and SecA1 when PsbO1 client protein is present. This would imply a ternary complex formation between HSP90C-SecA1-PsbO1. Prior studies on bacterial SecA have elucidated that the interaction between SecA and SecB is reinforced by the client protein itself [56]. Furthermore, the identification of overlapping binding interfaces on bacterial SecA, accommodating SecB and other ligands such as lipids, the translocon SecYEG, and the pre-protein, proves an indispensable role for the efficient transfer of pre-proteins from SecB to SecA and subsequent transport processes [57]. With the prokaryotic ancestry of chloroplasts, our inquiry aligns with established bacterial models of pre-protein translocation. We propose a collaborative interaction between plastid HSP90C and SecA1 in orchestrating pre-protein translocation at the SecYE channel, which is further enhanced by the client protein (Figure 5) [58].

## 4. Materials and Methods

### 4.1. Bimolecular Fluorescence Complementation (BiFC) and Transient Protein Expression Analysis

PsbO^1–58^, PsbO1, HSP90C, SecY1, SecA1, and RbcS3B coding regions were amplified and cloned into the pENTR207 vector by the Gateway system (Invitrogen). The BiFC destination vectors pB7WGYN9 or pB7WGYC9 containing yellow fluorescent protein (nYFP/cYFP) at their C-termini were prepared as previously described [59]. Plasmids used to overexpress SecA1-YFP, PsbO1^1–58^ YFP, PsbO1^1–85^ YFP, PsbO1^1–332^ YFP, and HSP90C-YFP were generated as described previously [24,26,60]. Plasmids for BiFC and transient expression analysis were transformed into Agrobacterium tumefaciens strain GV2260. *Nicotiana benthamiana* plants were grown for four weeks under long-day conditions (16 h of light, 8 h of darkness) at 28 °C. Full-length constructs were infiltrated at a final OD_600_ of 0.5, and the BiFC construct was infiltrated at a final OD600 of 0.2. Plants were imaged after 48 h by confocal microscopy. Before monitoring the fluorescence, 10 mM sodium azide or 30 μM geldanamycin was used to treat mesophyll cells after peeling the lower epidermis.

### 4.2. Fluorescence and Confocal Microscopy

Fluorescence microscopy was performed using an upright Zeiss LSM 510 confocal laser scanning microscope (Carl Zeiss). The excitation/emission wavelengths used for YFP signals are 440 nm/460–490 nm, and for chlorophyll, 633 nm/650–720 nm. Sequential z-stack scanning was used to avoid overlap between the fluorescence channels. Images were processed by ImageJ1.54f (National Institutes of Health, Bethesda, USA) or ZEN 2.3 (Carl Zeiss Microscopy GmbH, 2011, Oberkochen, Germany).

### 4.3. Protein Expression, Purification in E. coli, and Size Exclusion Chromatography Analysis

Constructs for His_6_-tagged HSP90C (His-HSP90C) and His_6_-tagged *m*PsbO1^T200A^ (PsbO1) were expressed in the pProEX HTb plasmid. The construct for His_6_-tagged *m*SecA1 was expressed in the pET28a plasmid. These constructs were all transformed into *E. coli* BL21 (DE3)-pRIL (Stratagene) carrying ampicillin or kanamycin resistance. For protein expressions, one liter of culture was generally grown for each construct in LB media with 100 μg/mL ampicillin until OD_600_ reached ~2.0. The cultures were incubated with 0.4 mM IPTG at 27 °C overnight. The induced cells were collected and lysed chemically with lysis buffer (25 mM Tris-HCl pH 7.5, 100 KCl, 5 mg/mL lysozyme, 1 mM DTT, 10% glycerol) and mechanically by sonication. Affinity purification using Ni-NTA resin (QIAGEN) was performed to isolate the His_6_-tagged proteins and they were further purified by size exclusion chromatography on the Superdex 200 10/300 GL column fitted in the fast protein liquid chromatography (FPLC) ÄKTApurifier 10 (GE Healthcare) system.

Size exclusion chromatography analyses were conducted on the same column. Protein mixtures were made using equal amounts (100 μg) and were incubated on ice for 1 h before being injected into the FPLC system. The mixture was allowed to flow through the column with a consistent flow of the isocratic elution buffer (25 mM Tris-HCl pH 7.5 and 100 mM NaCl). The eluted fractions were mixed with SDS sample buffer and analyzed on 10% SDS polyacrylamide gels. Gels were stained by either the standard silver staining protocol or the standard Coomassie Brilliant Blue staining protocol. Bands with about ~116 kDa apparent molecular weight for SecA1 were quantified using ImageJ (National Institutes of Health) and plotted using GraphPad Prism version 9.5.1 (GraphPad Software, Inc., Boston, MA, USA).

### 4.4. Steady-State ATP Hydrolysis Activity Assay

ATP-hydrolysis activity was measured using the coupled-NADH method as mentioned previously [43]. Purified HSP90C, SecA1, or PsbO1 proteins were mixed in equimolar compositions (1.41 μM) in the assay reaction mixture consisting of 25 HEPES (pH 7.5), 5 mM MgCl_2_, 500 mM KCl, 0.03% Tween 20, 10% glycerol, 200 μM NADH, 3 mM phosphoenol pyruvate, 15.7U of pyruvate kinase and 24.5U of lactate dehydrogenase (MilliporeSigma, Oakville, Canada). The extinction of NADH in the reaction mixture was mapped over time using the Synergy 4 microplate reader (BioTek, Winooski, USA). Wherever needed, radicicol and sodium azide were used to inhibit specifically the HSP90C and SecA1 ATPase activity, respectively. The turnover rate (*k*_cat_) and binding affinity (*K_M_*) were analyzed using the GraphPad Prism software.

### 4.5. In Vitro Chaperone Activity Assay

Citrate synthase (CS; Sigma) was first dialyzed in 20 mM HEPES/KOH (pH 7.5), 150 mM KCl, and 10 mM MgCl_2_. CS (1 μM) was prepared in a final reaction volume of 150 μL containing 20 mM HEPES/KOH (pH 7.5) and 2.8 mM β-mercaptoethanol with different amounts of purified HSP90C proteins. The mixtures were added to a 96-well microplate and heated at 45 °C. Light scattering at 340 nm was monitored at 45 °C in the Synergy 4 microplate reader. Control measurements were performed with purified HSP90 protein alone in the absence of CS.

### 4.6. Antibodies

Polyclonal rabbit anti-HSP90C antibody was described in [60]. Anti-PsbO1 antibody was generated by Signalway AntiBody (College Park, MD, USA) with purified His-PsbO1 protein. Anti-T7 antibody was purchased from Sigma-Aldrich. Monoclonal mouse anti-Lex A antibody was purchased from Santa-Cruz Biotechnology, and monoclonal anti-HA was purchased from Invitrogen.

## 5. Conclusions

In summary, the present investigation elucidates the interaction between the chloroplast HSP90 family protein, HSP90C, and SecA1 in vivo, revealing a dependence on specific intermediate conformations requisite for binding Sec translocon clients. The coordinated actions of ATP binding and hydrolysis emerge as pivotal determinants not only for facilitating SecA1-mediated protein transport through the Sec translocon but also for orchestrating interactions between HSP90C and its clients alongside SecA1. Furthermore, our findings underscore the dynamic nature of the Sec client transport process in vivo, highlighting the nuanced complexities of the intricate intermolecular associations, which may present challenges for faithful replication in vitro using purified proteins.

## Figures and Tables

**Figure 1 plants-13-01265-f001:**
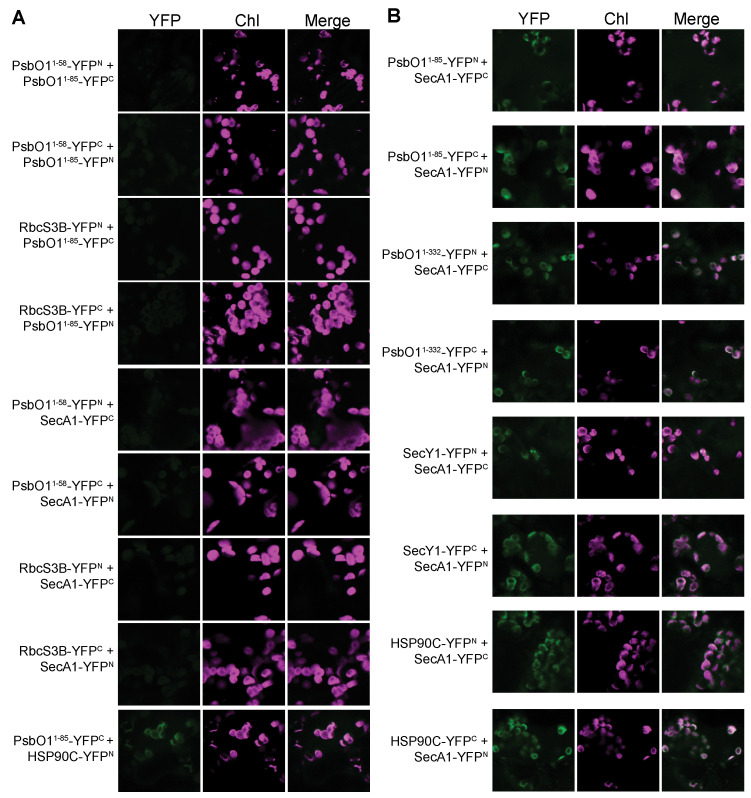
Bimolecular fluorescence complementation assays showing HSP90C interacts with SecA1 in vivo. Laser scanning microscopy images taken from mesophyll cells of *N. benthamiana* transiently co-expressing YFP N- or C-half-fusion proteins. Fluorescence images for YFP (YFP), chlorophyll (Chl), and the merged (Merge) signals are shown. Bars represent 5 μm. (**A**) Transient expression of PsbO1^1–85^-n/cYFP or SecA1-n/cYFP paired with stromal targeted controls (PsbO1^1–58^-c/nYFP or RbcS3B-c/nYFP). HSP90C and PsbO1^1–85^ pair was also tested. (**B**) Transient expression of Sec cargo (PsbO1^1–85^, PsbO1^1–332^)-n/cYFP paired with Sec translocon components (SecA1, SecY1)-c/nYFP, and transient expression of Sec motor (SecA1-n/cYFP) with HSP90C-c/nYFP.

**Figure 2 plants-13-01265-f002:**
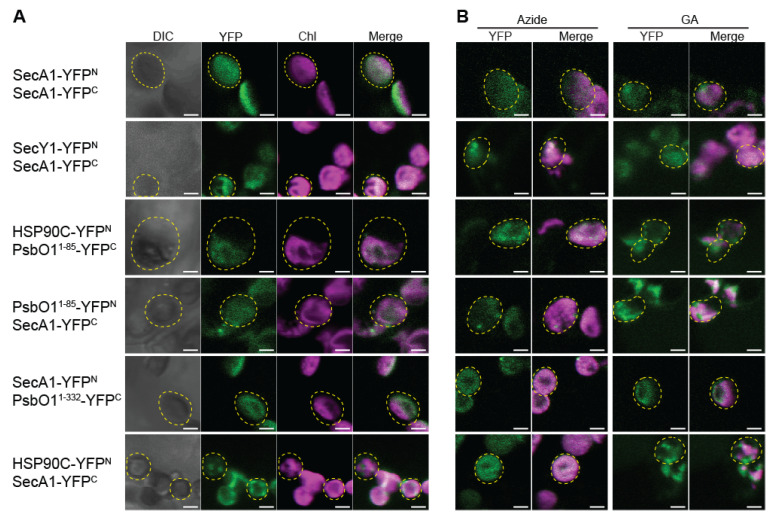
BiFC assays with SecA1 and HSP90C inhibitors reveal that the localization of interaction depends on the ATPase activity of both proteins. (**A**) Laser scanning microscopy images were taken of mesophyll cells of *N. benthamiana* transiently co-expressing YFP N- or C-half-fusion proteins that are labeled as -YFP^N^ and -YFP^C^, respectively. Fluorescence images for YFP (YFP), chlorophyll (Chl), and the merged (Merge) signals are shown. Bars represent 3.5 μm. (**B**) Mesophyll chloroplasts were incubated with 30 μM geldanamycin (GA) and 10 mM sodium azide (azide), respectively. Phase contrast (DIC) and fluorescence images for YFP, chlorophyll, and the merged signals are shown. Yellow dotted circles represent the border of chloroplasts. It should be noted the chloroplasts shown in (**B**) are not the same ones shown in (**A**).

**Figure 3 plants-13-01265-f003:**
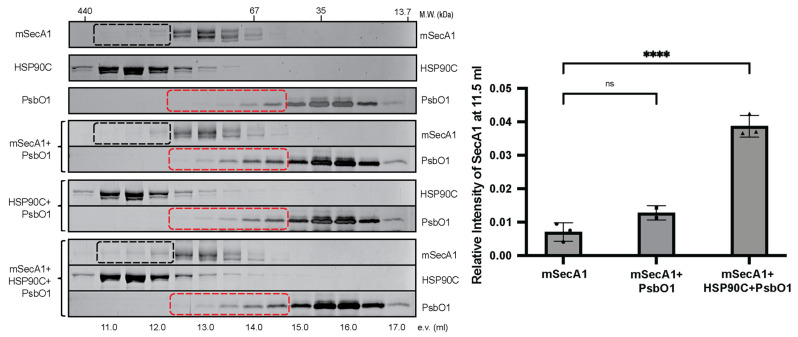
HSP90C and SecA1 interaction in vitro is enhanced by PsbO1 protein. Size exclusion chromatography analyses of SecA1, HSP90C, or PsbO1 alone or their mixtures. The proteins were resolved by 10% SDS-PAGE and stained by silver staining. Fractions corresponding to SecA1 and PsbO1, which correspond to potentially changed fractions, are highlighted with black and red rectangle boxes, respectively. The graph on the right shows the relative amount of SecA1 in fractions corresponding to 11.5 milliliters out of all fractions. Error bars reflect the standard deviation across three biological replicates, with asterisks denoting significance levels (****: *p* < 0.0001, ns: non-significant).

**Figure 4 plants-13-01265-f004:**
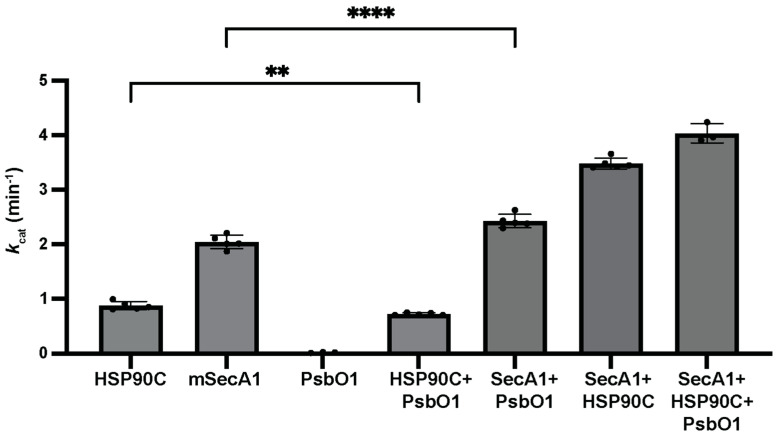
Influence of PsbO1 on HSP90C and SecA1 ATP hydrolysis activities. The *k*_cat_ values for HSP90C, SecA1 alone, and their equal molar mixtures (SecA1 + HSP90C), in the absence or presence of equal molar PsbO1. Error bars denote SD, with asterisks signifying significance levels (**: *p* < 0.005, ****: *p* < 0.0001).

**Figure 5 plants-13-01265-f005:**
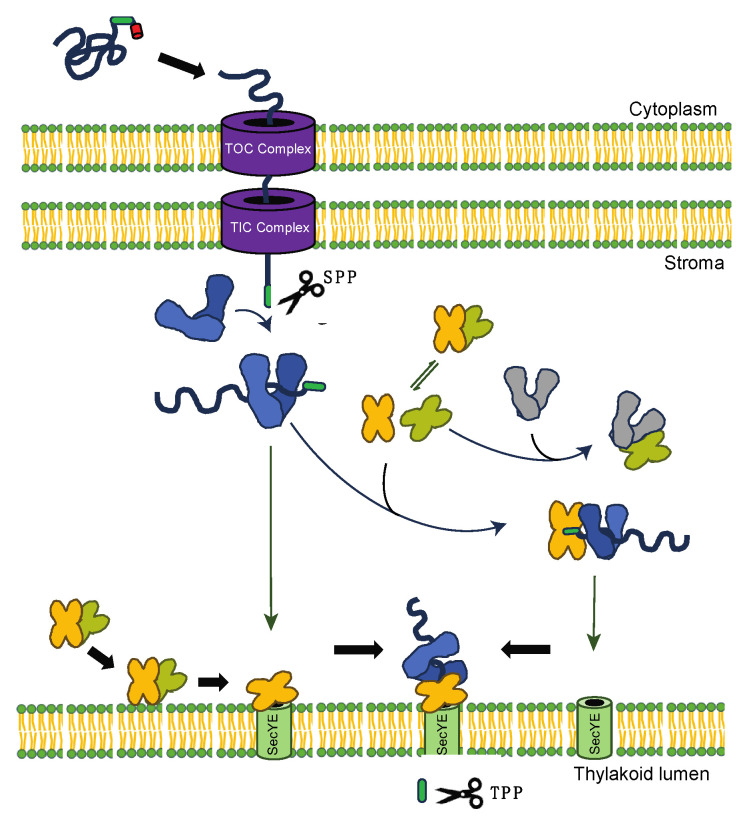
Proposed mechanism of HSP90C and SecA1 interaction during pre-protein translocation in plastid Sec1 system. Upon HSP90C-mediated (blue) stabilization of the Sec client protein, two pathways arise: one involving the complex interacting with dimeric SecA1 (yellow and green), corroborated by SEC analyses, and the other targeting pre-primed SecA1 directed to the SecYE channel.

## Data Availability

Primer sequences and all plasmid constructs are available upon request.

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
