# Peer review of "Plastid Molecular Chaperone HSP90C Interacts with the SecA1 Subunit of Sec Translocase for Thylakoid Protein Transport"

_plants, 2024, doi:10.3390/plants13091265_

Round 1

Reviewer 1 Report

Comments and Suggestions for Authors

In this manuscript the authors have examined the dynamic interplay between HSP90C, an essential plastid protein involved in chloroplast biogenesis and development, and SecA1 which functions in protein translocation across the thylakoid membrane. The authors show that in the presence of a client protein, in this case PsbO1, their interaction changes. Based on their results the authors propose a model which highlights the dynamic relationship between Hsp90C and SecA1 during protein translocation across the thylakoid membrane.  This is clearly a valuable study which provides new insights into chloroplast protein trafficking. However there are a few points the authors need to consider.

1.In Fig 1B I do not find that the distinction between stromal and thylakoid localization of the SecA1-YFP signal is convincing with  PsbO1 (1-85) and PsbO (1-332). More convincing data should be shown.

2.The authors had to use  a mutant PsbO1protein  to detect shifts in the elution profiles of SecA1 in combination with HSP90C and PsbO1 which suggest the formation of a complex. How can the authors exclude an artefact caused by the  mutation in PsbO1? This part of the manuscript could be strengthened by stabilizing the complex through crosslinking using WT PsbO1

Minor comments

In the Introduction the authors mention Jin et al 2022 for the TOC-TIC supercomplex. They also need to cite the study of Liu et al. 2023 published at the same time.

MBD needs to be defined in the text.

Author Response

In this manuscript the authors have examined the dynamic interplay between HSP90C, an essential plastid protein involved in chloroplast biogenesis and development, and SecA1 which functions in protein translocation across the thylakoid membrane. The authors show that in the presence of a client protein, in this case PsbO1, their interaction changes. Based on their results the authors propose a model which highlights the dynamic relationship between Hsp90C and SecA1 during protein translocation across the thylakoid membrane.  This is clearly a valuable study which provides new insights into chloroplast protein trafficking. However there are a few points the authors need to consider.

1.In Fig 1B I do not find that the distinction between stromal and thylakoid localization of the SecA1-YFP signal is convincing with  PsbO1 (1-85) and PsbO (1-332). More convincing data should be shown.

Reply: We really appreciate the reviewer’s comment that our study is valuable. In Figure 1B, we present the BiFC results displaying each test protein fused to half of the YFP molecule. When examining the interaction between SecA1 and PsbO11-85, as well as full-length PsbO11-332, we anticipate some similarities in their sub-organellar localizations. This expectation is confirmed by comparing the top two rows with the subsequent two rows. However, we observe a greater number of YFP signal spots not colocalized with the pink chlorophyll autofluorescence signal in the co-expression of SecA1 and PsbO11-332 half YFP fusion (third and fourth row), in contrast to the first two rows involving PsbO11-85. We also observe similar differences in the higher resolution images as shown in Figure 2A.

This difference strongly suggests a distinct subcellular localization of SecA1 interacting with the full-length PsbO1 protein within the stromal compartment. Furthermore, the SecA1-PsbO11-85 YFP co-expression pair, featuring solely the client protein's chloroplast targeting sequence and thylakoid signaling sequence, demonstrates an overlap between the pink chlorophyll autofluorescence signal and the green YFP signals, indicative of direct interaction between SecA1 and PsbO1 at the thylakoid membrane. This observation highlights a multi-stage interaction between SecA1 and PsbO1, which likely occurs sequentially. Specifically, the stromal interaction between SecA1 and the client protein likely arises due to steric hindrance imposed by the client protein itself. It is plausible that larger client proteins exhibit a preference for initial interaction with SecA1 within the stromal environment before translocating to the SecYE channel. Conversely, the heightened rate of thylakoid membrane interaction observed when utilizing solely the client protein's signal sequences suggests a preference for smaller or less sterically hindered client proteins to engage directly with SecA1 at the membrane interface. The spatial dynamics shown from these findings support a model wherein the interaction between SecA1 and PsbO1 occurs in discrete stages, with the stromal and membrane environments offering distinct modes of interaction dictated by the size and steric properties of the client protein.

2.The authors had to use a mutant PsbO1protein to detect shifts in the elution profiles of SecA1 in combination with HSP90C and PsbO1 which suggest the formation of a complex. How can the authors exclude an artefact caused by the  mutation in PsbO1? This part of the manuscript could be strengthened by stabilizing the complex through crosslinking using WT PsbO1

Reply: The PsbO1T200A mutation, characterized by a substitution of threonine at the 200th amino acid position with alanine, has been previously studied for its impact on protein function and in vivo thylakoid targeting. Investigations have corroborated the enhanced binding affinity of the PsbO1T200A mutant towards HSP90C, as validated by both yeast-two-hybrid (Y2H) assays and gel filtration experiments (Jiang et al., 2017). The mutant protein can effectively guide GFP protein for thylakoid transport leaving much less intermediate fusion protein in the stroma compared with the wild-type protein fusion (Jiang et al, 2020).  Consequently, our rationale for employing the mutant PsbO1 variant is to establish a stronger binary complex between HSP90C and PsbO1, with the anticipation that this enhanced interaction will facilitate improved binding to SecA1. Initially, attempts were made to utilize the WT PsbO1 protein in conjunction with HSP90C and SecA1 simultaneously. However, the limitations and detection thresholds associated with gel filtration assays rendered this approach challenging. Indeed, the formation and stabilization of a ternary complex in vitro present formidable hurdles, as it is contingent on specific protein conformations conducive to effective complex formation. Moreover, the technical complexities associated with the use of crosslinkers during gel filtration assays further compounded the experimental challenges. Furthermore, our paper stresses the critical role of protein dynamics, particularly in the context of SecA1 and HSP90C, in facilitating the formation of the ternary complex. Attempting to constrain these dynamic proteins to a singular conformation may prove counterproductive, potentially impeding the assembly of the desired ternary complex. While the proposed strategy holds merit, it is essential to consider the dynamic nature of the molecular assemblies involved in order to optimize the formation of the ternary complex effectively.

Minor comments

In the Introduction the authors mention Jin et al 2022 for the TOC-TIC supercomplex. They also need to cite the study of Liu et al. 2023 published at the same time.

Reply: Thank you for the comment. Yes, we added the reference in the manuscript as ref #22.

Jiang, T., Mu, B., & Zhao, R. (2020). Plastid chaperone HSP90C guides precursor proteins to the SEC translocase for thylakoid transport. Journal of Experimental Botany, 71(22), 7073–7087. https://doi.org/10.1093/jxb/eraa399

Jiang, T., Oh, E. S., Bonea, D., & Zhao, R. (2017). HSP90C interacts with PsbO1 and facilitates its thylakoid distribution from chloroplast stroma in Arabidopsis. PLoS ONE, 12(12), e0190168. https://doi.org/10.1371/journal.pone.0190168

Reviewer 2 Report

Comments and Suggestions for Authors

As a semi-autonomous organelle, chloroplasts have around 3000 proteins are translated in the cytosol and then translocated into chloroplasts. The whole translocation and sorting processes of these moved proteins is strictly regulated. In the traditional model of chloroplast protein translocation machinery, the complex containing multiple HSPs is considered functioning as motor proteins. HSPs are very abundant proteins in nature. In this study, the authors have conducted experiments to show the chaperon function of HSP90c and the interaction of HSP90c with SecA1 by the application of the client protein PsbO. In addition, the authors confirmed the motor activity of HSP90c by biochemical analyses. The MS was written well, and the introduction is comprehensive.

Unfortunately, the authors didn’t have a chance to show the chaperon activity of HSP90c, which is critical to understand the function of HSP90c. The holdase assay was strongly suggested to support their conclusions. In addition, Fig 3 panel A, it is hard to read the different shift of PSBO1 combined with SecA1 and HSP90c. Fig4, I am wondering why the analyses of the Influence of HSP90C on SecA1 ATP Hydrolysis Activities without the combination of three proteins.

Author Response

As a semi-autonomous organelle, chloroplasts have around 3000 proteins are translated in the cytosol and then translocated into chloroplasts. The whole translocation and sorting processes of these moved proteins is strictly regulated. In the traditional model of chloroplast protein translocation machinery, the complex containing multiple HSPs is considered functioning as motor proteins. HSPs are very abundant proteins in nature. In this study, the authors have conducted experiments to show the chaperon function of HSP90c and the interaction of HSP90c with SecA1 by the application of the client protein PsbO. In addition, the authors confirmed the motor activity of HSP90c by biochemical analyses. The MS was written well, and the introduction is comprehensive.

Unfortunately, the authors didn’t have a chance to show the chaperon activity of HSP90c, which is critical to understand the function of HSP90c. The holdase assay was strongly suggested to support their conclusions. In addition, Fig 3 panel A, it is hard to read the different shift of PSBO1 combined with SecA1 and HSP90c. Fig4, I am wondering why the analyses of the Influence of HSP90C on SecA1 ATP Hydrolysis Activities without the combination of three proteins. 

Reply: Thank you for your valuable input. We did investigate the chaperone activity of HSP0C in the lab but never published the work before. We have supplemented the manuscript with supplementary data pertaining to the general chaperone activity of HSP90C, as depicted in Figure S3a. This figure illustrates the ATP-independent chaperone activity of HSP90C, wherein we monitored its ability to mitigate thermal-induced citrate synthase aggregation over time. The results show a dosage-dependent inhibition of thermal-induced aggregation by HSP90C, indicating its pivotal role in maintaining protein homeostasis. Additionally, we validated the chaperone activity of the purified HSP90C, reaffirming its holdase activity in impeding the aggregation of aggregation-prone clients, consistent with the established paradigm of ATP-independent holdase activity exhibited by HSP90 ortholog (Daturpalli et al., 2013; Karagöz & Rüdiger, 2015). Furthermore, we observed the ATPase activity of HSP90C, wherein specific activity measurements elucidated an approximate Vmax value of 18.17 pmol/min and an ATP binding affinity (Km) value of 4.01 mM. These findings corroborate the canonical characteristics of Arabidopsis HSP90 ATPase activity observed across diverse organisms.

Employing gel filtration assays, we elucidate direct protein interactions. However, it became evident that the interactions detected via BiFC assays likely represent transient interactions, necessitating the involvement of additional cofactors to achieve the most optimal binding. Given the challenges associated with the detection limits of gel filtration assays, visualizing ternary complex formation in vitro using purified proteins remains technically challenging. Nevertheless, the observed shift represents a substantial deviation from baseline conditions, indicative of dynamic ternary complex formation occurring within the experimental conditions that we provided.

Our initial depiction of the turnover rate (kcat) value in Figure 4 for the HSP90C+SecA1 condition provided an estimated averaged kcat value for only one of the proteins. As requested by the reviewer, we now included the measured total turnover rate (kcat) for all three proteins together (last bar). Recognizing the shared characteristic of ATPase activity between the proteins within the experimental mixture, the revised Figure 4 now illustrates the combined kcat observed in the presence of both proteins and all three proteins. This adjustment ensures a more precise and comprehensive portrayal of the enzymatic kinetics involved. Upon analysis of the average turnover rates associated with the combined presence of HSP90C, SecA1, and PsbO1 proteins, we found complexities attributed to the distinct influences exerted by the client protein on SecA1 and on HSP90C. Consequently, the resulting alterations in turnover rates solely to either HSP90C or SecA1 remain unclear. Nevertheless, our data suggests a subtle enhancement in ATPase activity by PsbO1, relative to the SecA1+HSP90C condition, indicative of a potential stimulatory effect exerted, thus providing support for a ternary complex formation. However, elucidating whether both proteins collectively contribute to this stimulatory effect or if it is exclusively attributed to one of the proteins within the three-protein mixture necessitates further investigation.

References

 Daturpalli, S., Waudby, C. A., Meehan, S., & Jackson, S. E. (2013). Hsp90 Inhibits α-Synuclein Aggregation by Interacting with Soluble Oligomers. Journal of Molecular Biology, 425(22), 4614–4628. https://doi.org/10.1016/j.jmb.2013.08.006

Karagöz, G. E., & Rüdiger, S. G. D. (2015). Hsp90 interaction with clients. Trends in Biochemical Sciences, 40(2), 117–125. https://doi.org/10.1016/j.tibs.2014.12.002

Round 2

Reviewer 1 Report

Comments and Suggestions for Authors

The authors have satisfactorily answered my concerns.

Reviewer 2 Report

Comments and Suggestions for Authors

The authors have addressed the concerns I have, and they actually integreted the new results to support their conclusions. No I am fine with the acceptance of this MS and encourage to publish it with first priority.